# Candidate Acetic Acid Bacteria Strains for Levan Production

**DOI:** 10.3390/polym14102000

**Published:** 2022-05-13

**Authors:** Kavitha Anguluri, Salvatore La China, Marcello Brugnoli, Luciana De Vero, Andrea Pulvirenti, Stefano Cassanelli, Maria Gullo

**Affiliations:** Department of Life Sciences, University of Modena and Reggio Emilia, 42122 Reggio Emilia, Italy; kavitha.anguluri@unimore.it (K.A.); salvatore.lachina@unimore.it (S.L.C.); marcello.brugnoli@unimore.it (M.B.); luciana.devero@unimore.it (L.D.V.); andrea.pulvirenti@unimore.it (A.P.); stefano.cassanelli@unimore.it (S.C.)

**Keywords:** acetic acid bacteria, polysaccharides, levan, sucrose concentration

## Abstract

In this study, twelve strains of acetic acid bacteria (AAB) belonging to five different genera were tested for their ability to produce levan, at 70 and 250 g/L of sucrose concentration, respectively. The fructan produced by the bacterial strains was characterized as levan by NMR spectroscopy. Most of the strains produced levan, highlighting intra- and inter-species variability. High yield was observed for *Neoasaia chiangmaiensis* NBRC 101099 ^T^, *Kozakia baliensis* DSM 14400 ^T^ and *Gluconobacter cerinus* DSM 9533 ^T^ at 70 g/L of sucrose. A 12-fold increase was observed for *N. chiangmaiensis* NBRC 101099 ^T^ at 250 g/L of sucrose concentration. Levan production was found to be affected by glucose accumulation and pH reduction, especially in *Ko. baliensis* DSM 14400 ^T^. All the *Gluconobacter* strains showed a negative correlation with the increase in sucrose concentration. Among strains of *Komagataeibacter* genus, no clear effect of sucrose on levan yield was found. Results obtained in this study highlighted the differences in levan yield among AAB strains and showed interdependence between culture conditions, carbon source utilization, and time of incubation. On the contrary, the levan yield was not always related to the sucrose concentration.

## 1. Introduction

Levan or levan-type exopolysaccharides are non-structural homopolysaccharides formed by fructo-furanoses units. They consist of fructose monomers linked by β-(2,6) glycosidic bonds with possible β-(2,1) branches. Molecules are packed into nano-sized, spherical forms, which grant a very low intrinsic viscosity, high water solubility, and high water-retention capability (acting as hydrocolloids) [1,2,3]. Due to their chemical-physical characteristics, levan can be exploited in food (bulker, stabilizer, emulsifying, and encapsulating agent), cosmetics, biodegradable plastics, and detergents productions [2,4,5]. Moreover, their hydrolytic products (short-chain fructooligosaccharides) can stimulate the growth of some bifidobacteria in the colon, suggesting their prebiotic function [2,6,7,8]. Also, medical use of levan is reported, due to their ability to induce both strong interferon response and cell proliferation [9].

Levan is synthesized in the vacuole of some plant species usually with a low degree of polymerization or extracellularly by some *Archaea*, *Fungi*, and a wide range of bacteria (including AAB). Bacteria synthesize high molecular mass levan, having a high degree of polymerization and high intrinsic viscosity [6,10].

AAB are mainly known for the production of oxidation products, such as acetic acid, dihydroxyacetone and gluconic acid. Moreover, they are widely studied for their ability to synthesize exopolysaccharides, especially bacterial cellulose and levan [3,11,12,13,14,15]. In AAB, levan is synthesized in the periplasm by levansucrase enzyme (E.C. 2.4.1.10), also known as sucrose 6-fructosyltransferase, which has a high affinity to sucrose but low for fructose and glucose [16]. The reactions catalyzed by levansucrase are mentioned as the reversible primary step, where shuttling of fructosyl residue from carbonyl to carbonyl takes place, subsequent irreversible step in which transfer of fructosyl from carbonyl to carbinol and formation of levan occurs [17]. Active levansucrase has been found in both gram-positive (*Bacillus subtilis*, *Bacillus megaterium*, and *Streptococcus salivarius*) and gram-negative bacteria (*Zymomonas mobilis*, *Erwinia amylovora*, *Pseudomonas syringae*, and various AAB species) [18,19,20,21]. In gram-negative bacteria, like AAB, levansucrase is secreted out through a single peptide pathway, which helps the hydrolysis of sucrose and formation of levan by transfructosylation activity [22]. High concentration of substrate has been found to inhibit levansucrases in gram-negative bacteria because the interaction of enzyme-substrate occurs in the periplasmic space, resulting in the accumulation of enzyme and product. Whereas, in gram-positive bacteria, this is not observed, due to the presence of peptidoglycan wall [23].

Among AAB, levan production has been observed in different species of *Neoasaia (N.)*, *Kozakia (Ko.)*, *Komagataeibacter (K.), Gluconacetobacter (Ga.)* and *Gluconobacter (G.)* genera [8,24,25,26,27,28,29]. The first study on levan biosynthesis by AAB was conducted on the endophytic nitrogen-fixing bacterium, *Ga. diazotrophicus* SRT4 which produces a high amount of branched levan with a molecular weight above 2 × 106 Da [20]. Synthesis of levan was also confirmed for various strains of *K. xylinus* (*K. xylinus* NCI 1005, NCIM 2526) [8,25]. *N. chiangmaiensis* (=BCC 15763 ^T^ = NBRC 101099 ^T^), isolated from red ginger flowers, is known to produce levan-like exopolysaccharides (on sucrose deficient medium) and heteropolysaccharides into their environments [30,31]. Previously, Jakob and co-workers quantified and characterized the levan produced by strains of the species *G. frateurii*, *G. cerinus*, *N. chiangmaiensis*, and *Ko. baliensis* at 80 g/L of sucrose, by a combination of NMR and AF4-MALS-RI analysis [3]. These latter studies showed that the molecular weight of levan has high variability among AAB species, ranging from 4 MDa (*G. frateurii*) to 2000 MDa (*Ko. baliensis*). The molecular weight influences the physicochemical properties (such as different rheology) as well as the functioning (changes in antitumor and antiviral activities) of levan [32,33,34]. For those reasons, the high molecular weight levan are gaining great interest, especially in biomedical applications [35].

The main parameters affecting levan production by AAB are sucrose concentration, agitation speed, and media pH. Although the optimal sucrose concentration is a strain-dependent trait, the accumulation of glucose and carboxylic acid production (gluconic acid and acetic acid) must be considered for their contribution to enzyme inhibition and pH lowering [8,17,23]. Due to the acidic byproduct formation, the incubation time also plays an important role in obtaining high levan yield [29].

In this study, a pool of AAB strains was tested for levan production at two sucrose concentrations (70 g/L and 250 g/L). The effect of initial sucrose concentration, pH value, gluconic acid, and the residual carbon sources were analyzed. NMR analysis was conducted for levan characterization.

## 2. Materials and Methods

### 2.1. Microbial Strains

In this study, twelve AAB strains belonging to *Neoasaia*, *Kozakia*, *Gluconobacter*, *Komagataeibacter*, and *Acetobacter* genera were used (Table 1). The strains were collected from different culture collections, specifically UMCC (Unimore Microbial Culture Collection, Reggio Emilia, Italy), DSMZ (Deutsche Sammlung von Mikroorganismen und Zellkulturen, Braunschweig, Germany), NBRC (National Institute of Technology and Evaluation Biological Resource Center, Kisarazu, Japan), and ATCC (American Type Culture Collection, Middlesex, UK). Strains were cultured according to the indications of each culture collection.

### 2.2. Media and Culture Conditions

Culture media composition used in this study are reported in Table 2. Strain cultures recovered from −80 °C were initially refreshed in 5 mL of GYC broth [38] and incubated for 6 days at 28 °C in static conditions. Cultivation of strains for the production of levan was performed as described by Semjonovs et al., (2016) with some modifications [27]. Briefly, 2 mL of GYC broth culture was transferred in 13 mL Hestrin-Schramm (HS) medium and cultured for 3 days at 28 °C [39]. Precultures were prepared by inoculating 5 mL of former culture in 50 mL of HS, cultivated at 28 °C for 4 days in agitated conditions (100 rpm), using a rotary shaker incubator (Zhicheng ZHWY-200B, Shanghai, China). 

### 2.3. Levan Production

For producing levan, the HS medium was modified by replacing glucose with sucrose (HS-S), as a carbon source [27]. Cultivation was carried out by inoculating 0.5% of preculture in 40 mL of HS-S medium. Trials were performed at 70 and 250 g/L concentration of sucrose and incubated (30 °C) at agitation speeds 140 and 200 rpm, respectively, to facilitate the enzyme substrate interaction (sucrose-levanase) [40,41]. Assays were conducted in triplicate and samples, collected at 48 and 96 h of cultivation, were stored at −20 °C until further analysis.

### 2.4. Levan Extraction and Estimation

To determine levan concentration, crude polysaccharides were extracted from 4 mL of culture broth. To collect only polysaccharide fractions, excluding cell biomass and media components, ethanol precipitation was used as described by Semjonovs et al., (2016), with few modifications [27]. Briefly, the separation of cell biomass and polysaccharides was performed by centrifugation at 6000× *g* for 15 min. The resulting supernatant was added with 2.5 volumes of chilled absolute ethanol and stored at 4 °C for 24 h to allow the polysaccharide precipitation. Precipitated polysaccharides were collected by centrifugation at 6000× *g* for 15 min and dried at 60 °C. Pellets were freeze-dried (FREEZONE 1L Labconco, Kansas City, MO, USA) and stored for the quantification and determination of polysaccharides. The amount of levan was determined enzymatically by using the commercial fructan assay kit (K-FRUC^©^, Megazyme^®^, Bray, Ireland), following the manufacturer’s instructions.

### 2.5. NMR Spectroscopy

NMR spectra were acquired on a Bruker Avance Neo 400 spectrometer (Bruker BioSpin, Baden-Württemberg, Germany) operating at 100 MHz for ^13^C. Freeze-dried pellets of polysaccharides were dissolved in 600 µL of D_2_O. Levan control from the fructan assay kit (K-FRUC, Megazyme^®^, Bray, Ireland) and fructan extracted from *N. chiangmaiensis* 101099 ^T^ in this study were used as controls [3]. Chemical shifts were expressed in ppm relative to the internal standard trimethylsilane (TMS).

### 2.6. Determination of Sucrose, Sugar Monomer Residues, and Gluconic Acid

For the determination of the residual carbon sources, broth of strain cultures grown on HS-S was collected from three biological replicates at 48 and 96 h, respectively. Samples were centrifuged at 16,000× *g* for 5 min to remove cell biomass and any residual low weight molecules, as suggested by [42]. Enzymatic analysis was performed on the supernatant to estimate the concentration of residual sugars (sucrose, glucose, and fructose) using the commercial K-SUFRG enzymatic kit (Megazyme^®^, Bray, Ireland), and gluconic acid using K-GATE kit (Megazyme^®^, Bray, Ireland), previously suggested by [16,42,43]. pH and titratable acidity were measured by using an automatic titrator (Titroline^®^ easy, Schott, Mainz, Germany), samples were neutralized by titrating against NaOH (1 M) to pH 7.0.

### 2.7. Statistical Methods

All the experiments were conducted in triplicates. Data were represented as the average of the triplicate ± standard error. The statistical significance was determined using two-way ANOVA and the statistical differences among samples were assessed using Tukey post-hoc test. The correlation of sucrose concentrations and levan yield was calculated based on the Pearson coefficient, indicating a positive correlation R^2^ ≥ 0.6 or negative correlation for R^2^ ≤ −0.6. All statistical tests were performed using R, Version 4.1.0.

## 3. Results

### 3.1. Effect of Sucrose Concentration on Levan Yield

The production kinetic of levan, assayed at 70 g/L and 250 g/L of sucrose, is shown in Figure 1. The effect of sucrose concentration on levan yield was evaluated by calculating the Pearson coefficient, which was expressed as −0.6 ≥ R^2^ ≥ 0.6, as negative and positive correlation, respectively (Appendix A) [44]. The production trends of levan showed high variability among the tested strains and was not always strictly correlated with the increase of sucrose concentration. The highest levan yield was obtained from *N. chiangmaiensis* NBRC 101099 ^T^. At 250 g/L of sucrose, the higher amount of levan was produced during 48 h (43.5 g/100 g), and further increased at 96 h reaching 63.21 g/100 g. When a lower amount of sucrose (70 g/L) was supplied, the levan production trend was completely different compared to 250 g/L, showing a high levan production at 48 h (4.81 g/100 g) and a slight increase at 96 h (5.2 g/100 g; R^2^ = 0.65, *p* = 0.003534). Levan production for *Ko. baliensis* DSM 14400^T^ showed a different tendency compared to *N. chiangmaiensis* NBRC 101099 ^T^. In both sucrose conditions, levan production was characterized by an increase at 48 h, followed by a marked reduction at 96 h. As described for *N. chiangmaiensis* NBRC 101099 ^T^, sucrose concentration positively impacts the levan yield (34.99 g/L at 250 g/L of sucrose; R^2^ = 0.63; *p* = 0.005).

A completely different scenario was depicted for the three *Gluconobacter* strains (*G. cerinus* DSM 9533 ^T^, *G*. *frateurii* DSM 7146 ^T^, and *G. oxydans* DSM 2343) regarding the levan production among conditions. These strains showed a consistent levan production at 48 h, when cultured at 70 g/L of sucrose (5.67 g/100 g for *G. cerinus* DSM 9533 ^T^; 2.64 g/100 g for *G. frateurii* DSM 7146 ^T^ and 3.26 g/100 g for *G. oxydans* DSM 2343), with no further increase until 96 h. A negative correlation between sucrose concentration and levan yield (Appendix A) was observed for *Gluconobacter* strains suggesting that high sucrose concentration negatively affects levan production. Among strains of the *Komagataeibacter* genus, of which two *K. hansenii* (*K. hansenii* DSM 5602 ^T^ and *K. hansenii* ATCC 53582) and three *K. xylinus* (*K. xylinus* K2G30, *K. xylinus* DSM 6513 ^T^ and *K. xylinus* DSM 2004), there was no clear effect of sucrose concentration on levan yield (−0.6 < R^2^ < 0.6). Two strains among *K. xylinus (K. xylinus* DSM 2004 and *K. xylinus* K2G30) produced a considerable yield of levan, showing a positive effect of sucrose concentration on levan production. Differences among the two strains in the production tendency are related to the cultivation time. *K. xylinus* DSM 2004 showed an increase of the production for all the period (reaching 31.38 g/100 g at 250 g/L of sucrose). In contrast, *K. xylinus* K2G30 showed a similar phenotype of *Ko. baliensis* DSM 14400 ^T^, characterized by a consistent production at 48 h and a marked reduction of levan yield at 96 h in 250 g/L of sucrose. Regarding *K. hansenii*, both strains were negatively affected by sucrose concentration. Our dataset also included two strains of *Acetobacter* genus (*A. pasteurianus* AB0220, and *A. pasteurianus* DSM 3509 ^T^) of which, only *A. pasteurianus* AB0220 was able to produce a considerable amount of levan (16.88 g/100 g/ 96 h) at 250 g/L of sucrose. The two former strains showed no correlation between levan production and sucrose concentration (R^2^ < 0.6).

### 3.2. Gluconic Acid Formation and pH Reduction

Organic acids formation during cultivation results in pH fluctuations, which negatively affect levansucrase activity [26]. Based on the medium composition, gluconic acid could be produced from the glucose accumulation. In Figure 2a,b, pH variation according to gluconic acid production was reported, as an average of three replicates (± standard error). Change in the pH value was observed at 48 h of incubation, and further reduction to 2.8–3.5 was observed in five strain cultures (*N. chiangmaiensis* NBRC 101099 ^T^, *Ko*. *baliensis* DSM 14400 ^T^, *G. cerinus* DSM 9533 ^T^, *G. frateurii* DSM 7146 ^T^, and *K*. *xylinus* K2G30) when cultured at 70 g/L of sucrose. High variability in the production of gluconic acid was observed in some strains (e.g., *N. chiangmaiensis* NBRC 101099 ^T^ and *Ko. baliensis* DSM 14400 ^T^), showing similar pH values. The lowest pH among the tested strains was reached in *Ko. baliensis* DSM 14400 ^T^ at 96 h, along with high titratable acidity values (Appendix A), giving a putative explanation of the reduction of levan yield between 48 and 96 h. Similar pH were observed in *G. cerinus* DSM 9533 ^T^ and *G. frateurii* DSM 7146 ^T^, although they show variation in gluconic acid production. When cultured at 250 g/L, *Gluconobacter* strains showed no variations in pH and consequently the production of gluconic acid was nearly 0. *K. xylinus* K2G30 showed similar pH fluctuations to *N. chiangmaiensis* NBRC 101099 ^T^ at 70 g/L, coupled to a high amount of gluconic acid produced. Among the two tested *Acetobacter* strains, *A. pasteurianus* AB0220 showed rapid drop in pH from 48 to 96 h when cultured at 250 g/L (from 6.4 ± 0.35 to 3.56 ± 0.30), due to a higher production of gluconic acid (5.83 ± 0.56 at 96 h).

### 3.3. Carbon Sources Utilization and Consumption during Levan Production

Based on the medium composition, sucrose will be used as sole carbon source. Levan sucrase activity will hydrolyze sucrose into fructose and glucose: fructose will be used for levan production by levansucrase, while glucose will be used by the cells for the basal metabolism and for the production of other compounds. High sucrose consumption (Figure 3a) was observed in *N. chiangmaiensis* 101099 ^T^ cultured at 250 g/L of sucrose (206.94 ± 3.36 g/L at 96 h). Residual glucose (Figure 3b) and fructose (Figure 3c), resulting from sucrose hydrolysis, showed variability in consumption according to the initial sucrose amount supplied. At 70 g/L, *N. chiangmaiensis* 101099 ^T^ consumed glucose and fructose parallelly, whereas at 250 g/L fructose was rapidly consumed, and accumulation of glucose was observed. The behavior of *Ko*. *baliensis* DSM 14400 ^T^ in terms of sucrose concentration was slightly different from *N. chiangmaiensis* 101099 ^T^, which showed maximum consumption of sucrose in the first 48 h (63.1 ± 3.4 (90%) at 70 g/L and 180.43 ± 38.5 g/L (72%) at 250 g/L). Moreover, glucose utilization was higher in *Ko*. *baliensis* DSM 14400^T^ at 250 g/L (60.33 ± 6.21 g/L at 96 h) compared to *N. chiangmaiensis* 101099 ^T^ (37.95 ± 3.12 at 96 h). All *Gluconobacter* strains showed a similar trend of the carbon sources utilization (sucrose, glucose, and fructose) at 70 g/L of sucrose, they consumed the carbon sources in the first 48 h and remained unchanged until the end of 96 h of incubation. At 250 g/L of sucrose, a limited consumption of carbon sources was observed in *Gluconobacter* strains. Among *Komagataeibacter* strains, *K. xylinus* DSM 2004 showed high consumption of sucrose when cultured at 250 g/L of sucrose. Similarly, at 250 g/L condition, *A. pasteurianus* AB0220 consumed a high amount of sucrose in the first 48 h, while glucose, and fructose utilization were high at 96 h (32.26 g/L and 36.53 g/L, respectively).

### 3.4. NMR Analysis for Levan Characterization

The anomeric configurations of levan were analyzed by using ^13^C NMR spectra on samples collected from 250 g/L condition at 96 h of incubation. Strains of 5 different genera, namely *Neoasaia*, *Kozakia*, *Komagataeibacter, Acetobacter*, and *Gluconobacter* were used for this analysis. Previous studies on the carbon chemical shift values, related to levan, were used as a reference for the comparison of the tested samples [3,45,46].

The ^13^C NMR spectrum depicted six main resonances for carbon signals in all the tested strains (C1–C6). The resonance represents one quaternary carbon (δ 104.12–104.23 ppm; C-2) and five signals in 58–81 ppm range (Figure 4), where C2 and C6 correspond to the β-(2–6)–linkages The position of the carbon signals from *N. chiangmaiensis* NBRC 101099 ^T^, *Ko. baliensis* DSM 14400 ^T^, *K. hansenii* ATCC 53582, and *A. pasteurianus* DSM AB0220 (Table 3) were identical with those levan produced by *Zymomonas mobilis*, *Tanticharoenia sakaeratensis* [46], *Bacillus subtilis* MTCC 441 [47]. *G. cerinus* DSM 9533 ^T^ was the only exception showing 12 carbon signals attributable to sucrose structure (α-D-glucopyranosyl-(1→2)-β-D-fructofuranoside) [45,48,49]. This could be due to the low amount/quantity of fructan produced by *G. cerinus* DSM 9533 ^T^, which could not be detected. Moreover, chemical shifts of C2 and C6 which correspond to the β-(2–6)–linkages were in accordance with the controls and previous literature confirming the presence of levan [3,50].

## 4. Discussion

In this study, a total of 12 strains belonging to 5 different genera of AAB (*Neoasaia*, *Kozakia*, *Komagataeibacter*, *Acetobacter*, and *Gluconobacter*) were tested for levan production. Most of the strains tested produced levan in the conditions tested, which was characterized by NMR. Variability in the production of levan, highlighted that for some of them (*N. chiangmaiensis* NBRC 101099 ^T^, *Ko. baliensis* DSM 14400 ^T^, *K. xylinus* strains and *A. pasteurianus* AB0220) the suitable sucrose concentration is 250 g/L. The remaining strains, including those of *Gluconobacter* genus and *K. hansenii*, were more productive at 70 g/L of sucrose, showing no or very low production of levan at 250 g/L. Based on the yield obtained, the best levan producer was *N. chiangmaiensis* NBRC 101099 ^T^, followed by *Ko. baliensis* DSM 14400 ^T^, *K. xylinus* DSM 2004 and *A. pasteurianus* AB0220.

Previous studies reported high variability among strains in sucrose utilization for levan production [29,36,46]. Low levan production at 50 g/L of sucrose, that increased at 100 g/L of sucrose, was previously observed [29]. Higher substrate concentration (250 g/L of sucrose) was already tested using AAB, maximizing the levan yield and defining the highest sucrose concentration suitable for levan production [46]. Maximum yield of levan at 80 g/L of sucrose was found in *Ga*. *diazotrophicus*, whereas 250 g/L of sucrose were optimal for levan production by *Tanticharoenia sakaeratensis* [24,29,46,51].

Strains tested in our study belong to species known to possess abundant levansucrase copy numbers, which can justify the high levan yield [25,27,28]. The sucrose consumption resulted to be independent among the higher producing strains, highlighting the influence of other parameters than sucrose concentration on levan production.

pH fluctuation, due to organic acids production, influences levan yield and its molecular weight. Previous studies reported that pH variation resulting from gluconic and acetic acid production [29], or glucose accumulation, as reported by González-Garcinuño et al. [23] affect levan production. According to the culture conditions assayed in this study, the expected organic acid was gluconic acid, which is produced from glucose oxidation.

Data of this study, confirm the influence of pH on levan yield, except for one of the tested strains (*N. chiangmaiensis* NBRC 101099 ^T^), although a reduction of pH was observed, reaching 2.8, no effect was observed on levan yield. The same phenotypic behavior was observed when *N. chiangmaiensis* NBRC 101099 ^T^ was cultured in sourdough in which a rapid drop of pH was observed without affecting levan yield [29]. Other strains were influenced by pH fluctuation. This was highly evident in the case of *Ko. baliensis* DSM 14400 ^T^, for which we observed a drastic reduction of levan production after 48 h of cultivation. The decrease of levan yield was accompanied by gluconic acid production and glucose depletion. This was in accordance with previous studies, in which the maximum yield was observed in the first 35 h of cultivation, followed by a slight degradation until 48 h [29]. Based on this evidence, a definition of the harvesting time could be considered as a parameter for setting the best levan yield. Furthermore, the accumulation of glucose and sucrose in the medium could negatively affect levan yield by activating the levanase transcription, an enzyme responsible for the hydrolysis of levan, contrasting the levansucrase activity [6,52]. In our study, levanase activity was not assayed, however the reduction of levan yield accompanied by an increase of fructose concentration could be attributed to hydrolysis of levan by the levanase activity.

In this study, high variability in levan production among tested AAB was observed. The initial sucrose concentration, which is considered as a limiting factor for producing levan, does not always affect levan yield. This phenotypic behavior was not a species-specific trait. The best strain tested in this study for levan production was *N. chiangmaiensis* NBRC 101099 ^T^. Considering the cultivation time, *Ko. baliensis* DSM 14400 ^T^ achieved the highest levan yield in the first 48 h of cultivation. Our results also suggest the suitability of *Komagataeibacter* strains for levan production, in particular *K. xylinus* DSM 2004 and *K. xylinus* K2G30, and the strain *A. pasteurianus* AB0220. Future studies should be addressed aimed at characterizing genes involved in levan synthesis and kinetics parameters related to enzymes of the levan synthesis pathway.

## Figures and Tables

**Figure 1 polymers-14-02000-f001:**
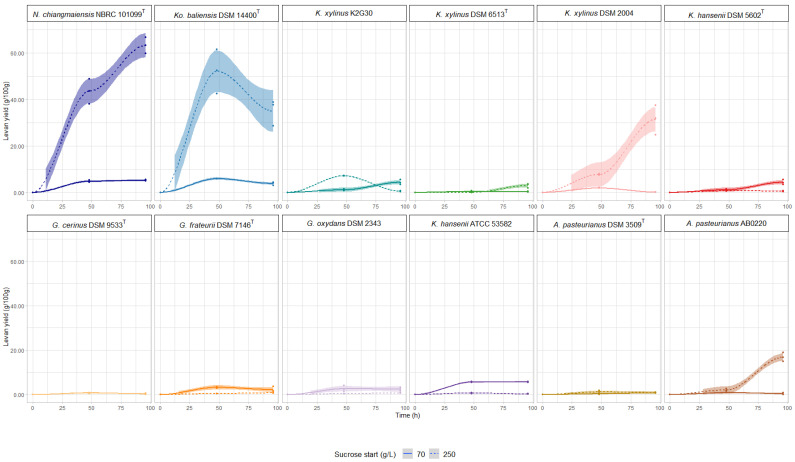
Change in levan production based on sucrose concentration. Values are represented with a confidence interval (CI) of 95%.

**Figure 2 polymers-14-02000-f002:**
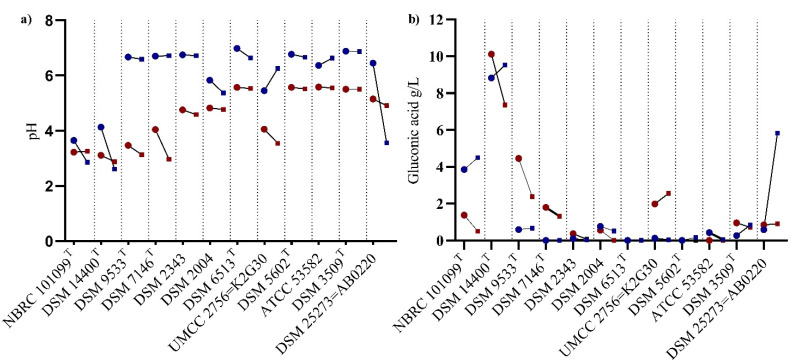
(**a**) pH variation at 70 g/L and 250 g/L sucrose concentrations, at 48 and 96 h of incubation (vs) (**b**) Production of gluconic acid under similar conditions. 
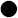
—48 h of incubation; 
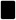
—96 h of incubation; 

—color refers to 250 g/L sucrose); 

—color refers to 70 g/L sucrose.

**Figure 3 polymers-14-02000-f003:**
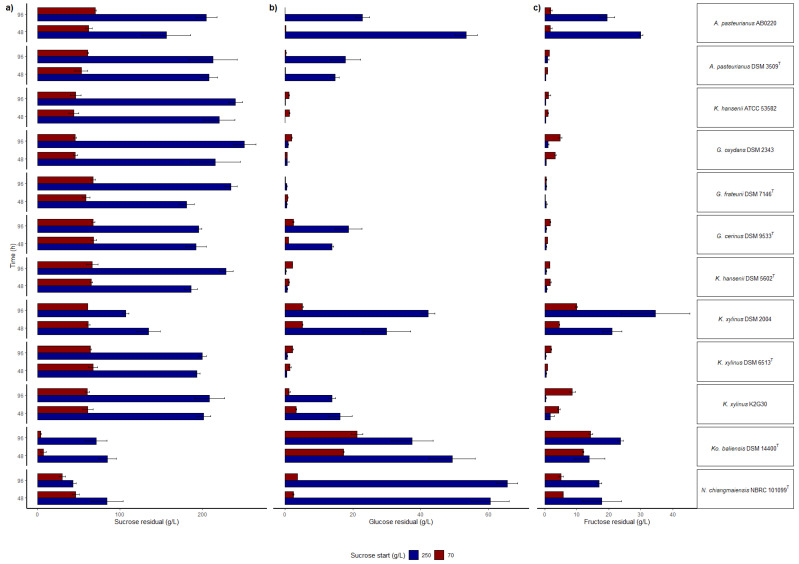
Residual carbon sources, (**a**) sucrose, (**b**) glucose, and (**c**) fructose, in the culture medium at 48 and 96 h of incubation. Bar plot indicates the average consumption by three replicates ± standard error.

**Figure 4 polymers-14-02000-f004:**
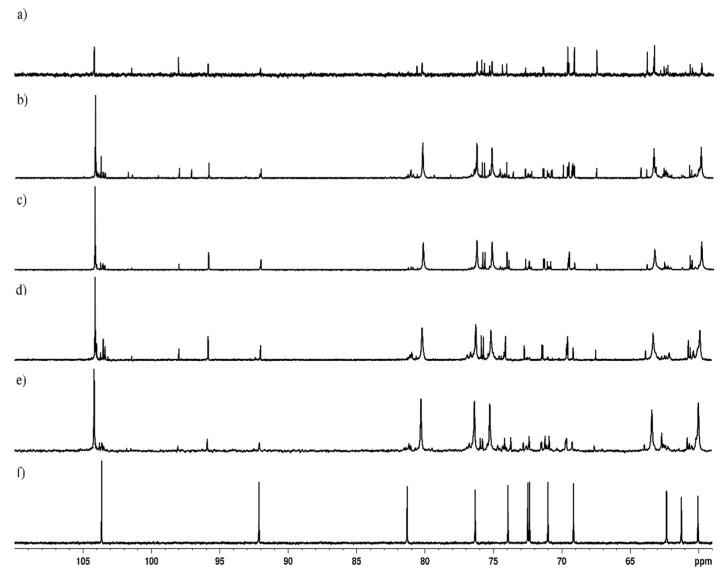
^13^C NMR spectra of levan produced by AAB strains cultured at 250 g/L sucrose concentration; (**a**) Levan control flour (Megazyme^®^) (**b**) *N. chiangmaiensis* NBRC 101099 ^T^; (**c**) *Ko. baliensis* DSM 14400 ^T^; (**d**) *K. hansenii* ATCC 53582; (**e**) *A. pasteurianus* AB0220; (**f**) *G. cerinus* DSM 9533 ^T^. α-D-glucopyranosyl-(1→2)-β-D-fructofuranoside highlights the detection of sucrose.

**Table 1 polymers-14-02000-t001:** List of AAB strains tested in this study.

Strain	Collection ID	* Sucrose (*w*/*v*)	Reference
*N. chiangmaiensis*	NBRC 101099 ^T^	80/50-100	[29,36]
*Ko. baliensis*	DSM 14400 ^T^	80/50–100	[29,36]
*G. cerinus*	DSM 9533 ^T^	80	[36]
*G. frateurii*	DSM 7146 ^T^	80	[36]
*G. oxydans*	DSM 2343	80	[36]
*K. xylinus*	DSM 2004	-	-
*K. xylinus*	DSM 6513 ^T^	-	-
*K. xylinus*	UMCC 2756 = K2G30	-	-
*** K. hansenii*	DSM 5602 ^T^	80	[36]
*** K. hansenii*	ATCC 53582	-	-
*A. pasteurianus*	DSM 3509 ^T^	-	-
*A. pasteurianus*	UMCC 1754 = AB0220	-	-

* Referred to tested sucrose concentration in previous studies; ** Recently *K. hansenii* was reclassified into *Novacetimonas* genus [37].

**Table 2 polymers-14-02000-t002:** Composition of media used in this study.

Component (% *w*/*v*)	Medium		
	GYC	HS	HS-Sucrose (HS-S)
Glucose	10	2	-
Sucrose	-	-	7/25
Yeast extract	1	0.5	0.5
Poly-peptone	-	0.5	0.5
Na_2_HPO_4_	-	0.73	0.73
Citric acid	-	0.115	0.115
MgSO_4_	-	0.05	0.05
CaCO_3_	2	-	-
Agar	1.5	-	-

**Table 3 polymers-14-02000-t003:** Chemical shifts of ^13^C NMR for characterization of levan produced from AAB strains.

Strain	Chemical Shifts (ppm) Carbon Atoms					
	C1	C2	C3	C4	C5	C6
Levan control flour (Megazyme)	59.89	104.21	76.26	75.16	80.26	63.33
*N. chiangmaiensis* NBRC 101099^T^	59.89	104.12	76.29	75.16	80.26	63.36
*Ko. baliensis* DSM 14400 ^T^	59.91	104.16	76.32	75.18	80.21	63.33
*K. hansenii* ATCC 53582	59.89	104.12	76.26	75.16	80.23	63.26
*A. pasteurianus* UMCC 1754 = AB0220	60.0	104.23	76.38	75.18	80.44	63.35
*G. cerinus* DSM 9533 ^T^	92.16	71.10	72.51	69.22	72.38	60.15
	61.39	103.81	76.35	74.01	81.42	62.38

## Data Availability

Not applicable.

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
