# Peer review of "Candidate Acetic Acid Bacteria Strains for Levan Production"

_polymers, 2022, doi:10.3390/polym14102000_

Round 1

Reviewer 1 Report

This paper titled  “Candidate acetic acid bacteria strains for levan production” is interesting. This manuscript could be considered for publication in Polymers after major revising.

My comments are as follow:

  1. The author needs to check the whole text including grammar, punctuation, writing, et al. There are too many errors.
  2. The objectives of this study should be summarized in the abstract.
  3. Please compare your data with previous studies in the discussion section.

Reviewer 2 Report

Comments from the reviewer

The problem of searching for new highly productive bacterial strains to obtain levan polysaccharide is important and relevant. However, the manuscript needs to be improved.

Some comments are:

  1. Pg 3 , line 94-98. Chapter 2.2. Media and culture conditions and Pg 3, line 106-110. Chapter 2.3. Levan production. The description of culture conditions is not clear. Was this procedure described in the literature? If so, please add the reference.
  1. Why was the HS medium chosen for obtaining levan, which is used to obtain BC? Was this procedure described in the literature? If so, please add the reference.
  2. Why the authors used the following processing parameters: amount of sucrose 70 g/L and 250 g/L? Its discussion is required.

Pg 3, line 103-106. The authors write that at these concentrations of sucrose the highest yield of levan is observed and refer to works in which completely different bacteria Zymomonas mobilis and B. subtilis were used. At the same time, the authors themselves write that the optimal sucrose concentration is a strain-dependent trait.

  1. Pg 6, line 184-185. «The initial pH of culture media was set between 5.6 and 7 at sucrose concentrations of 70 and 250 g/L, respectively». This is a non-specific and incomprehensible sentence.
  2. Pg6Chapter 3.2. Gluconic acid formation and pH reduction. Why do you associate a decrease in pH with the formation of gluconic acid, and not other acids? It is necessary to prove that it is gluconic acid.
  3. Pg 10, line 308. The authors write that «This study provides the basic data in understanding the levan production in AAB». In our opinion, this is too self-confident statement.
  4. Start the discussion with a paragraph that contains the main findings of the paper.
  5. Since the title of the article is «Candidate acetic acid bacteria strains for levan production», it should be clearly indicated in the conclusion which AAB species and strains the authors consider the most promising for levan production.
  6. To assess the concentration of residual sugars (sucrose, glucose and fructose), an enzymatic identification method was used using a commercial set of enzymatic k-SUFRG (Megazyme ® , Bray, Ireland) corresponding to the manufacturer's analysis method (https://www.megazyme.com/documents/Assay_Protocol/K-SUFRG_DATA.pdf). At the same time, the methodology indicates that glucose is measured before and after sucrose hydrolysis, and fructose is measured after phosphoglucose isomerase isomerization. But the culture fluid may contain its own enzymes that hydrolyze sucrose, including in the measurement process, and NADPH may also be present. In addition, the methodology states that "The assays are specific for D-glucose and D-fructose. Since β-fructosidase also hydrolyzes low-molecular-weight fructans (for example, kestose), this method, like all others, is not absolutely specific for sucrose. Some indication of the presence of fructooligosaccharides will be given by the ratio of D-glucose to D-fructose when determined after hydrolysis by β-fructosidase. A deviation from 1:1 (an increase in the proportion of D-fructose) would indicate the presence of fructan. This can be verified by measuring D-fructose in a “sucrose sample" after determining total D-glucose." All this may introduce an error in the measurement or a more detailed presentation of the methodology is required.

Round 2

Reviewer 1 Report

The authors addressed all my comments.

The manuscript may need further format according to MDPI requirements. 

Reviewer 2 Report

The authors have taken into account most corrections and recommendations from the reviewers.